# A Synchronous Prediction Model Based on Multi-Channel CNN with Moving Window for Coal and Electricity Consumption in Cement Calcination Process

**DOI:** 10.3390/s21134284

**Published:** 2021-06-23

**Authors:** Xin Shi, Gaolu Huang, Xiaochen Hao, Yue Yang, Ze Li

**Affiliations:** School of Electrical Engineering, Yanshan University, 438 Hebei Avenue, Qinhuangdao 066004, China; shixin0329@163.com (X.S.); huanggaolu2018@sina.com (G.H.); 18712737027@163.com (Y.Y.); lize_1995@163.com (Z.L.)

**Keywords:** energy consumption prediction, moving window, multi-channel convolutional neural networks

## Abstract

The precision and reliability of the synchronous prediction of multi energy consumption indicators such as electricity and coal consumption are important for the production optimization of industrial processes (e.g., in the cement industry) due to the deficiency of the coupling relationship of the two indicators while forecasting separately. However, the time lags, coupling, and uncertainties of production variables lead to the difficulty of multi-indicator synchronous prediction. In this paper, a data driven forecast approach combining moving window and multi-channel convolutional neural networks (MWMC-CNN) was proposed to predict electricity and coal consumption synchronously, in which the moving window was designed to extract the time-varying delay feature of the time series data to overcome its impact on energy consumption prediction, and the multi-channel structure was designed to reduce the impact of the redundant parameters between weakly correlated variables of energy prediction. The experimental results implemented by the actual raw data of the cement plant demonstrate that the proposed MWMC-CNN structure has a better performance than without the combination structure of the moving window multi-channel with convolutional neural network.

## 1. Introduction

The cement industry is considered as an energy intensive sector [1]. The energy consumption of the entire cement manufacturing process depends largely on the electricity and coal consumption [2]. Accurate and timely energy consumption prediction is of great significance for reasonable energy scheduling, energy saving, and a reduction in production costs [3,4]. As one of the most important processes of cement production, the cement calcination process is to calcine ground raw material into cement clinker. Traditionally, electricity and coal consumption are measured mainly by sensors and weighing machines, respectively, however, the changing trend and coupling relationship of electricity and coal consumption in cement production cannot be detected, which result in the faultiness in providing guidance to energy scheduling and production optimization. Due to the lack of a coupling relationship between electricity and coal consumption, the optimal solution for the optimal control of the production process may not be in the solution space, which hinders the monitoring and optimization of the cement manufacturing process and the formulation of a process control strategy. Consequently, it is necessary to investigate a method that can realize the synchronous prediction of electricity and coal consumption.

Accurate and reliable synchronous prediction of multi energy consumption indicators is complicated by the fact that the cement calcination process consists of multiple production process variables. The relationships between cement production process variables and energy consumption indicators have the characteristics of time-varying delay, uncertainty, and nonlinearity. For example, as the amount of raw materials increases, the electricity and coal consumption required for the calcination process increase. It takes 50–60 min, varying with the different conditions, for the cement raw material to be converted into calcined clinker. As a result, the electricity and coal consumption required for the calcination of raw materials in the rotary kiln cannot be calculated using the current amount of raw materials, and can only be manually adjusted by the operators based on historical experience. Using the current number of process variables to predict the energy consumption indexes will result in data misalignment and low accuracy. Therefore, the impact of the time-varying delay on the prediction of electricity and coal consumption of the cement calcination process needs to be considered. In production, a large amount of production process data are recorded, which makes it possible to synchronously predict electricity and coal consumption by data driven methods.

On the basis of the collection of numerous industrial data and the development of data driven methods, various data driven forecasting approaches have been proposed for prediction or diagnosis in the industrial field. For example, fault detection using statistical regression [5], electricity system load forecasting [6], and tendency prediction of the blast furnace hearth thermal state [7] using support vector machine (SVM) and electricity consumption prediction [8]. Furthermore, in the cement manufacturing industry, power consumption is predicted by the multiple non-linear regression algorithm [9] and empirical mode decomposition based hybrid ensemble model [10].

However, the above investigations all focused on a single index prediction strategy, which is insufficient for coal and electricity synchronous prediction in the cement calcination process because the coupling relationship of electricity and coal consumption cannot be obtained. The optimal control solution of the cement calcination process obtained from single index prediction model may not be in the actual solution space, which is invalid as it cannot be used to optimize the cement calcination process.

Although many multi-task process studies have been done in many fields such as the forecasting of the maximum connections in wireless communication [11], the prediction of solar radiation with multi-time scale and multi-component [12], quality monitoring of wind turbine blade icing processes [13], multi-task-oriented production layout in manufacturing factories [14], and multi objective task scheduling in cloud computing [15], there are still obstacles caused by time-varying delay and parameter redundancy. Time-varying delay between variables is considered to be one of the characteristics of process industry. It is difficult to build accurate prediction models without considering the time delay problem. The operating conditions are constantly changing in actual cement production, which causes the time delay between the process variables and the target variables, which vary with the process time. Thus, the timing matching of process variables and target predictors loses its effect due to the time-varying delay. The parameter redundancy is another feature. Synchronous prediction models have some advantages that make them indispensable. However, in the simultaneous prediction of multi-energy indicators of the cement calcination process, there are weak correlations between some input process variables, which have impacts on the prediction accuracy when simultaneously used as input data.

According to the above analyses, a multi-indicator predication model was established to simultaneously predict the coal and electricity consumption in the cement calcination process. On one hand, this reduces the modeling burden in the practical application of industrial systems, which contributes to the efficient use of industrial system resources. On the other hand, it provides a precise prediction of coal and electricity consumption in the cement calcination process, which contributes to the reasonable production scheduling and energy planning of the cement calcination process. In addition, the multi-output prediction model can well explain the strong coupling relationship between the prediction indicators, which is consistent with the process mechanism of the cement calcination process. Therefore, research into this issue is of great significance to promote the development of the cement industry.

The proposed model combines moving window with convolutional neural networks (CNN) to reduce the negative impact of time-varying delay on the prediction of cement energy consumption. The moving window integrates time-varying delay information hidden in time series data into the input layer of the CNN. As a result of powerful feature extraction capability, CNN is not only used in image processing [16], but also in manufacturing industry [17], which is used to extract the data characteristics of variables in the cement calcination process.

The multi-channel structure of CNN was designed to reduce the negative impact of the redundant parameters of weakly correlated variables on cement energy consumption prediction in which the single index prediction strategy was first used to predict the electricity and coal consumption separately, with the purpose of reducing the coupling relationship, and then, the coupling relationship was rebuilt before the final output in order to fuse the variables’ characteristics. The design of a multi-channel structure in this paper not only preserves the practical application advantages of multi-output models, but also reduces the negative impact of parameter redundancy on the energy consumption prediction.

The combination of a moving window, multi-channel structure with CNN contributes the two characteristics of this paper:

(1) A method combining moving window and multi-channel convolutional neural networks is proposed to accommodate time-varying delay information implied in the time series data according to the analysis of the mechanism of cement calcination. The time-varying delay is a kind of unique phenomenon in process industries such as the cement industry. The variable data in the previous period are accommodated by moving window, which is more efficient for feature extraction by CNN with a one-dimensional convolution kernel. The adoption of CNN effectively reduces the excessive parameters brought by moving windows and avoids complex timing matching.

(2) The multi-channel structure of MWMC-CNN was designed to predict coal and electricity consumption simultaneously, which effectively addresses the parameter redundancy problem caused by correlated variables of the cement calcination process. In addition, the adverse impact on energy consumption indicators of the synchronous prediction caused by the coupling relationship of the cement production process variables is also eliminated. Finally, the coupling relationship of electricity and coal consumption was rebuilt to provide references for the optimization of the cement calcination process.

The actual raw data of the cement plant were used to train the model and the experimental results demonstrate its superiority.

The rest of this paper is organized as follows. In Section 2, we introduce the related works about the existing literature related to data driven forecasting. Section 3 describes the technological process of a cement rotary kiln and the selection of prediction variables. The proposed MWMC-CNN is detailed in Section 4. The models with different structures were simulated, and the prediction results compared with least square support vector machine (LSSVM), CNN, and long short term memory (LSTM) in Section 5. Our conclusions and future research directions are given in Section 6.

## 2. Related Work

Due to the extensive usage of sensors, a large number of industrial data are recorded, which creates the basis for the optimization and operational control of the system by data driven methods. There have been many data driven prediction investigations for energy consumption based on statistical regression, SVM, tree-based method, ANN, and LSTM.

Statistical regression is a direct method for data driven, which is a predictive model technique that studies the relationship between independent variables and dependent variables, and is widely used for short-term predictions in many industries. Bianco applied linear regression to predict urban power consumption [18] and Catalina applied it to forecast the thermal energy consumption of buildings [19]. The advantage of linear regression is that the structure is simple, but its nonlinear ability is limited and it is easily interfered by outliers. Compared with linear regression, principal component analysis and partial least squares regression analysis have faster speed, better nonlinear ability, and a good effect in the short-term load forecasting of the power system [20]. Although the methods of statistical regression are simple and easy to implement, they are susceptible to outliers, however, while dealing with complex problems, they are also prone to overfitting. In addition, statistical regression methods cannot eliminate the negative impact of time-varying delays on prediction accuracy.

SVM is a kind of typical machine learning method that has strong nonlinear ability, and the computational complexity depends on the number of support vectors, so it still has better performance even if the sample space has a higher dimension [21]. These advantages show that it can be used to conduct energy analysis by continuous time series data in industrial fields. Based on SVM, Wang achieved the prediction of hydropower consumption [22] and Assouline estimated the solar photovoltaic potential [23]. Furthermore, some investigations were implemented to optimize SVM, where the optimal parameters of each kernel function [24] and optimization algorithms were obtained [25]. Although SVM can deal with the nonlinear regression problem and its parameter can be adjusted by the corresponding adaptive algorithm, the performance of SVM is still limited when the amount of training data is large [26]. The scale of time series data with time-varying delays recorded from the production line is usually very large, thus support vector machine cannot perform well when dealing with these data.

Tree-based methods divide the predictor space into sub-regions, and then the predicted results based on statistical indicators are obtained, which can be used not only for classification but also regression. Tree-based models are widely used to predict critical bus voltage and load in PV buses [27], the heat load of residential buildings [28], and the hourly performance forecasting of ground source heat pump systems [29]. The use of tree-based methods in industry have also been investigated by scholars such as the directed edge weight prediction of industrial Internet of Things [30] and the analysis of industrial data in cognitive manufacturing [31]. Regression trees split the training dataset into distinct and non-overlapping regions, which is inappropriate for energy consumption forecasting in cement manufacturing because of the strong coupling in cement production data.

ANN is a data-driven method of artificial intelligence that has a complex structure and is used for information processing problems that cannot be solved by theoretical analysis [32]. It is applied to industrial process analysis because of the advantages of adaptive learning, fast optimization, and strong nonlinear ability [33]. ANN has been used to forecast electricity consumption [34,35] annually or household electricity consumption daily and hourly [36] as well as day-ahead electricity price [37]. Moreover, it is also used for the prediction of Polish gas consumption [38]. Although the above studies fully utilized the advantages of artificial neural networks, appropriately selected the variables associated with the target variables, established a model for the corresponding actual situation, and achieved certain results, ANN takes a set of handcrafted features as the input and it is difficult to obtain the most useful features relevant to the estimation task [39].

LSTM is good at processing time series prediction because of its special structure, which is relevant to time factor. Thus, LSTM is applied to predict energy consumption, for example, gas consumption [40]. Besides, it is widely utilized for electricity consumption prediction such as the energy consumption of housing [41] and commercial buildings [42], medium- and long-term power forecasting [43], and electricity load forecasting in the electric power system [44]. Although all of the above investigations achieved the prediction of electricity consumption and are used as references for electric departments or companies for decision-making in power production and dispatching, there is only one forecasting index that is discrepant with the purpose of optimizing the industrial production process.

Deep learning not only shows excellent performance in image analysis [45], speech recognition [46], and text understanding [47], but it is also applied in the industrial field [48], which inspires us to apply deep learning to predict electricity and coal consumption synchronously.

CNN is a popular structure of deep learning, which was first proposed by Y. LeCun et al. (1998) [49]. The characteristics of weight sharing, local connection, and pooling reduction parameters give it computational advantages in the processing of big data [50] and additional feature engineering research on data is unnecessary [51], thus some scholars have applied CNN to industrial analysis such as big data classification of the Internet of Things [52] and civil infrastructure crack defect detection [53]. Since the inputs of the above two studies are two-dimensional data, which is different to one-dimensional time series data, the difference may cause the loss of the advantages of one-dimensional time series data [54]. In response to this situation, convolutional neural networks with one-dimensional convolution kernels are used in industrial fields such as industrial machine health status detection [55] and fault diagnosis for the condition monitoring of gearbox [56]. In the above two studies, the CNN with one-dimensional convolution kernel is more effective than the traditional structure in the extraction of the characteristics of time series data, which inspired this paper because of the nonlinear and non-stationary cement prediction energy synchronous production.

## 3. Process Analysis and Variable Selection of Cement Industry

The cement calcination system is the main energy consumption subsystem in the cement production process. Aiming at this subsystem, the relevant variables reflecting energy consumption and the time range of the moving window were selected for the proposed method. This section describes the mechanism of the cement calcination process and the reasons for selecting the parameters required for the model.

The cement calcination process can be roughly divided into three parts: preheating and pre-decomposition, calcination, and cooling, during which the cement raw material is converted into calcined clinker. The cement calcination process is shown in Figure 1.

First, the cement raw material is sent into the cyclone preheater to preheating. The cement raw material in the preheater is in suspension due to the action of the ID fan, which increases the contact area between the gas and raw material. The high temperature gas is discharged from the rotary kiln and the decomposition furnace exchanges heat with the cement raw material sufficiently, and the larger contact area contributes to the decomposition of carbonate in the raw material. Second, the cement raw material is sent into the rotary kiln that rotates at a constant speed for calcination, which contributes to the rapid decomposition of the carbonate of the cement raw material and the progress of other physical and chemical reactions in the rotary kiln. Finally, the high temperature clinker from the kiln is cooled by the grate cooler until the temperature can be withstood by a subsequent process. It takes about 50–60 min for the cement raw material to be converted into calcined clinker. Therefore, the electricity consumption of per ton cement production (ECPC) and the coal consumption of per ton cement production (CCPC) of the current batch are reflected by the production process variables in the past hour.

As shown in Figure 1, the main energy consumption of the cement calcination process consists of coal consumption and electricity consumption; different energy consumptions are marked by different color fonts. A large amount of coal is consumed to maintain the high temperature in the entire cement rotary kiln and the preheater of the decomposition furnace. The cement rotary kiln flips a lot of raw materials and the ID fan transports the high temperature gas, so these processes consume a lot of electric energy. Furthermore, the decomposition furnace and the rotary kiln both consume a large amount of coal to maintain the temperature. Therefore, some production process variables affect both the consumption of electricity and coal, which causes an uncertain coupling relationship between electricity consumption and coal consumption. In addition, the energy consumption of cement calcination systems in the future is related to the working conditions of the past, which reflects the energy consumption of the past period, affecting that in the future.

From the above analysis, we can see that the electricity and coal consumption of the cement calcination process are affected by the amount of cement raw material and the energy consumption of the past period. Therefore, the amount of cement raw material, the ECPC, and CCPC at historical moments were selected for the prediction of coal and electricity consumption. In addition, the coal consumption is impacted by the temperature in the preheater, decomposition furnace, and rotary kiln, so the process variables selected for coal consumption prediction were as follows: primary cylinder temperature, decomposition furnace coal consumption, decomposition furnace temperature, kiln temperature, secondary air temperature, and kiln head coal consumption. Electricity consumption is influenced by process variables such as kiln current and fan speed, so the ID fan speed, EP fan speed, Kiln average current were selected for the prediction of electricity consumption.

## 4. The Establishment of the MWMC-CNN Model

According to the above analysis, time-varying delay, parameter redundancy, and coupling are the main barriers in the forecasting of coal and electricity consumption simultaneously, thus we propose a MWMC-CNN structure that employs a multi-channel CNN model and moving window technique to solve the time-varying delay and data redundancy problem of energy consumption prediction in the cement calcination process. The moving window technique was employed to establish the input layer of the MWMC-CNN model to solve the time-varying delay problem. The multi-channel structure was adopted in the proposed model instead of the traditional structure to solve the parameter redundancy problem. This section describes the components of the proposed MWMC-CNN model and the specific steps of the energy prediction algorithm for the cement calcination process.

### 4.1. The Structure of the MWMC-CNN Model

The structure of the proposed model can be divided into three parts. The first part is a time-varying delay data input layer, where the cement calcination process variables are processed in time series. The second part is the data feature extraction layers in which different cement calcination process variables are convolved and pooled separately. The third part is a regression prediction layer that integrates the output data of two independent channels and finally outputs the predicted energy consumption value. Then, the Adam algorithm propagates back errors that come from the output values of MWMC-CNN and the training tags to layers of MWMC-CNN, and then modifies the weights until the errors are less than the expected values. The structure of the MWMC-CNN model is shown in Figure 2.

#### 4.1.1. The Time-Varying Delay Input Layer Structure of MWMC-CNN

The moving window technique was employed to establish the input layer of the MWMC-CNN model to solve the time-varying delay problem. The establishment can be divided into three parts: variables selection, window parameter selection, and time series moving window construction.

The first step in this section is to select the key variables that affect the energy consumption of the cement calcination process and normalize the selected variables data $\hat{X}$ as follows:(1)$$X=\frac{\hat{X}-{\hat{X}}_{\min}}{{\hat{X}}_{\max}-{\hat{X}}_{\min}}$$

According to the analysis of the energy consumption of the cement calcination process in Section 3, the amount of raw material ($X_{1}$), the ECPC at historical moments ($X_{2}$), and the CCPC at historical moments ($X_{3}$) affect the coal and electricity consumption in future. Therefore, they were selected as input variables in both channels.

The selected six variables for the A channel were as follows: ID fan speed ($X_{A_{1}}$), kiln average current ($X_{A_{2}}$), EP fan speed ($X_{A_{3}}$), the amount of raw material ($X_{1}$), ECPC at historical moments ($X_{2}$), and CCPC at historical moments ($X_{3}$). As for the B channel, the selected nine variables were as follows: primary cylinder temperature ($X_{B_{1}}$), decomposition furnace coal consumption ($X_{B_{2}}$), decomposition furnace temperature ($X_{B_{3}}$), kiln temperature ($X_{B_{4}}$), secondary air temperature ($X_{B_{5}}$), kiln head coal consumption ($X_{B_{6}}$), the amount of raw material ($X_{1}$), ECPC at historical moments ($X_{2}$), and CCPC at historical moments ($X_{3}$).

Second, the time intervals of the data selection window are selected. As shown in Figure 3, the time interval of selecting data was selected to be *s*, which means that the time series data of the cement production process variables that contain the time-varying delay characteristic are sent to the input layer. The variable data of past *s* time stamps correspond to the predicted value of energy consumption indicators after *p* time stamps, which means that the *t−s~t* period variable data in the $\hat{X}$ are adopted, to predict the energy consumption indicators at *t* + *p* time stamps. The size of the moving window should be larger than the interval of time-varying delay, which means that the time delay information contained in the time series data can be extracted by the model to eliminate its influence on prediction accuracy.

Finally, the data selection window is continuously sliding at unit time on the input time series data to realize the construction of the time series moving window. As shown in Figure 3, different rows represent different variables of the *A* input channel. During the model training process, the cement calcination process variables data are continuously selected by the rolling data selection window and there are always *m* groups of data, which are arranged into a matrix in the input sequence. For instance, the amount of raw material ($X_{1}$) selected by the moving window is expressed as
(2)$$x_{1}=(X_{1}(t-s+1),X_{1}(t-s+2),\cdot \cdot \cdot ,X_{1}(t))$$

The other variables selected are in the same form as Equation (2).

The data selected as the *A* input channel by the time series moving window are as follows:(3)$$x^{\left(s\times 6\right)}={\left(x_{A_{1}},x_{A_{2}},x_{A_{3}},x_{1},x_{2},x_{3}\right)}^{T}$$
where $x^{\left(s\times 6\right)}$ represents the input matrix of *A* channel with *s* rows and *6* columns, which means that the input sequence contains *6* variables in the past *s* sampling times.

The data selected as the *B* input channel by the time series moving window are as follows:(4)$$x^{\left(s\times 9\right)}={\left(x_{B_{1}},x_{B_{2}},x_{B_{3}},x_{B_{4}},x_{B_{5}},x_{B_{6}},x_{1},x_{2},x_{3}\right)}^{T}$$
where $x^{\left(s\times 9\right)}$ represents the input matrix of *B* channel with *s* rows and *9* columns, which means that the input sequence contains *9* variables in the past *s* sampling times.

#### 4.1.2. The Structure of the MWMC-CNN Data Feature Extraction Layer

Weak correlation exists among some input variables, which reduces the prediction accuracy. The multi-channel structure of CNN was designed in the data feature extraction layers of the proposed model to solve the parameter redundancy problem between weakly correlated variables.

As shown in Figure 4, the variable data of the cement calcination process selected by the moving window become the input time series data. Then, the time series data enter the data feature extraction layer of the proposed model, which performs convolution and pooling operations on the selected time series data through two independent channels. The *A* channel is used to extract the time series information about electricity consumption during the cement calcination process, and the *B* channel is used for the coal consumption.

Convolution on time series data using one-dimensional convolution kernels is considered to be efficient. The one-dimensional convolution kernel $w_{u}$, rather than square, is used by the proposed model to extract the time delay information implied in the time series data of the cement calcination process. The size of the one-dimensional convolution kernel is h×1, and the convolution stride is 1, which means that the kernel with length h moves by one step every time. For the A channel, the input time series data are convoluted by n one-dimensional convolution kernels as follows:(5)$$a_{i}^{\left((s-h+1)\times 6\right)}=\sigma_{relu}(w_{i}x^{\left(s\times 6\right)}+b_{i}),i=1,2,\cdot \cdot \cdot n$$
where $w_{i}$ represents the convolution kernel; the biases of *w_i_* is $b_{i}∙x^{\left(s\times 6\right)}$, which represents the input time series data with dimension of s×6, meaning that the input data has s rows and 6 columns. $a_{i}^{\left((s-h+1)\times 6\right)}$ represents the data after they are convolved and activated. $\sigma_{relu}(.)$ represents the relu activation function, which is defined as follows:(6)$$\sigma_{relu}(x)=\max(0,x)$$

The input time series data are activated by the *relu* function after being convolved by *n* convolution kernels, and *n* feature maps are output as the input of the pooling layer. Every feature map is equivalent to a data matrix. To compress the characteristics of feature maps, the one-dimensional pooling kernels were used as follows:(7)$$q_{i}^{\left(((s-h+1)/k)\times 6\right)}=f_{\mathrm{avg}-pooling}(a_{i}^{\left((s-h+1)\times 6\right)}),i=1,2,\cdot \cdot \cdot n$$
where $a_{i}^{\left((s-h+1)\times 6\right)}$ represents the output of convolution layer. $q_{i}^{\left(((s-h+1)/k)\times 6\right)}$ represents the data after pooled by the one-dimensional pooling kernels. The proposed model adopts one-dimensional average pooling as follows:(8)$$f_{\mathrm{avg}-pooling}(x)=\frac{1}{k}\sum_{\tau =1}^{k}x_{\tau }$$

One-dimensional average pooling takes an average for every *k* value in the column and the pooling stride is *k*. In the proposed method, the dimension of input feature maps is (s−h+1)×6 and the dimension of output feature maps is ((s−h+1)/k)×6, which means that the time series data are compressed vertically by the one-dimensional pooling kernel with a size of k×1 and stride of *k*. The length of feature map matrices is reduced while the number of which is unchanged after the one-dimensional pooling.

The above processes are one convolution layer and one pooling layer of the *A* channel, which is constructed by repeating the processes multiple times, and the structure of the *B* channel is similar to *A* except for the size of the input data matrix.

#### 4.1.3. The Regression Prediction Layer of MWMC-CNN

Although the interference problem of weakly correlated variables does not exist in the regression prediction layer, it exists in the process of convolution and pooling, thus the multi-channel structure was adopted by the proposed model to solve this problem. The fully connected layer integrates the time-varying delay features extracted from the two CNN channels, rebuilding the coupling relationship of coal and electricity consumption. Therefore, the time-series data that have been convolved and pooled multiple times in each channel are integrated into a column as the input of the fully connected layer. As shown in Figure 2, the A channel outputs $T_{1}$ neurons, the B channel outputs $T_{2}$ neurons, thus the fully connected layer is inputted $T_{1}+T_{2}$ neurons. Each input data of the fully connected layer is connected to the T neurons, which are also the output units of the fully connected layer, thus every neuron of the fully connected layer has $T_{1}+T_{2}$ weights. The outputs of the full connection layer are as follows:(9)$$y_{z}=\sigma_{relu}(w_{z}\rho_{u}+b_{z}),z=1,2,\cdot \cdot \cdot ,T,u=1,2,\cdot \cdot \cdot ,T_{1}+T_{2}$$
where $\rho_{u}$ is the input data of fully connected layer; $w_{z}$ represents the weight of the fully connected layer neuron; $b_{z}$ represents the bias corresponding to $w_{z}$; and $y_{z}$ represents the output of the fully connected layer neurons.

The output data of fully connected layer are linearly weighted and summed to obtain the outputs of the proposed model as follows:(10)$$y_{\alpha }=\sum_{z=1}^{T}\left(y_{z}\right),\alpha =1,2$$
where $y_{\alpha }$ represents the predicted values of the proposed model; $y_{1}$ represents the ECPC of the cement calcination process; $y_{2}$ represents the CCPC of the cement calcination process; and $y_{z}$ represents the output data of neurons of the fully connected layer.

#### 4.1.4. Parameter Adjustment Algorithm of MWMC-CNN

Traditional CNN usually uses the stochastic gradient descent method for back propagation, but it is prone to overfitting and local optimal solutions. To prevent these disadvantages, it can be replaced by the Adam algorithm [57], which iteratively updates the neural network weights based on training data. Adam designs independent adaptive learning rates for parameters *w* and *b* by calculating the first moment estimation and second moment estimation of the gradient. The Adam algorithm is used to perform backpropagation to adjust the parameters by inversely adjusting the weights *w* and the biases *b* of the multi-channel convolutional neural network model proposed in this paper. For the regression prediction problem, it is convenient to use the mean square error as the objective function to measure the gradient increases or decreases with error, which contribute to the convergence of the model. The study of this paper is the synchronous regression prediction problem of coal and electricity consumption in the cement calcination process. The *MSE* (mean squared error) is performed as the objective function of the proposed model, which is implemented as follows:(11)$$Minimize:J(w,b;x)=\sum_{i=1}^{n}{\left(y_{\alpha }(w,b;x)-\hat{y}\right)}^{2}$$
(12)$$x={\left[\begin{array}{cccc} X_{A_{1}}(t-s+1) & X_{A_{1}}(t-s+2) & \cdots & X_{A_{1}}(t)\\ X_{A_{2}}(t-a+1) & X_{A_{2}}(t-a+2) & \cdots & X_{A_{2}}(t)\\ \vdots & \vdots & \cdots & \vdots \\ X_{B_{6}}(t-a+1) & X_{B_{6}}(t-a+2) & \cdots & X_{B_{6}}(t) \end{array}\right]}^{T}$$
where $\hat{y}$ represents the true value of energy consumption in cement calcination process; $y_{\alpha }(w,b;x)$ represents the predicted energy consumption of the proposed model; and x represents the time series data of the cement calcination process, which are input into the A channel and B channel of the proposed model to predict ECPC and CCPC, respectively.

The following process is the update process of weights *w* and bias *b* by the Adam algorithm. The default parameters of the Adam algorithm are preserved, which include the learning rate *r* (r=0.001), the exponential decay rate of the moment estimate $\beta_{1}$ and $\beta_{2}$ ($\beta_{1}=0.9$, $\beta_{2}=0.999$); and the constant δ ($\delta =10^{-8}$).

First and second moment variables and time steps are initialized to be zero ($m_{0}=0$, $v_{0}=0$, t=0). The error converges to a certain value ε (ε=0.001), which is regarded as the stopping criterion of the proposed model. In the training process, time steps increase step by step (t=t+1) and in every step, *L* sets of data are randomly selected from the cement calcination process variable training dataset as follows:(13)$$x_{random}^{L}=\{x^{\left(1\right)},x^{\left(2\right)},\cdots ,x^{\left(l\right)},\cdots ,x^{\left(L\right)}\}$$
where $x^{\left(l\right)}$ represents a randomly selected set of training samples. The average gradient $g_{w_{t}}$ and $g_{b_{t}}$ at time step *t* of the *L* samples are calculated according to the adopted objective function as follows:(14)$$g_{w_{t}}=\frac{1}{L}\nabla_{w_{t}}\sum_{l=1}^{L}J(w_{t-1};x^{\left(l\right)})$$
(15)$$g_{b_{t}}=\frac{1}{L}\nabla_{b_{t}}\sum_{l=1}^{L}J(b_{t-1};x^{\left(l\right)})$$

The biased first moment estimates based on the gradient are updated as follows:(16)$$m_{w_{t}}=\beta_{1}\cdot m_{w_{\left(t-1\right)}}+(1-\beta_{1})\cdot g_{w_{t}}$$
(17)$$m_{b_{t}}=\beta_{1}\cdot m_{b_{\left(t-1\right)}}+(1-\beta_{1})\cdot g_{b_{t}}$$

The biased second moment estimates based on the gradient are updated as follows:(18)$$v_{w_{t}}=\beta_{2}\cdot v_{w_{\left(t-1\right)}}+(1-\beta_{2})\cdot g_{w_{t}}$$
(19)$$v_{b_{t}}=\beta_{2}\cdot v_{b_{\left(t-1\right)}}+(1-\beta_{2})\cdot g_{b_{t}}$$

The deviations of the first moment estimates based on the biased first moment estimate are corrected as follows:(20)$${\hat{m}}_{w_{t}}=m_{w_{t}}/(1-\beta_{1}^{t})$$
(21)$${\hat{m}}_{b_{t}}=m_{b_{t}}/(1-\beta_{1}^{t})$$

The deviations of the second moment estimates based on the biased second moment estimates are corrected as follows:(22)$${\hat{v}}_{w_{t}}=v_{w_{t}}/(1-\beta_{2}^{t})$$
(23)$${\hat{v}}_{b_{t}}=v_{b_{t}}/(1-\beta_{2}^{t})$$
where $\beta_{1}^{t}$ and $\beta_{2}^{t}$ represent the values of $\beta_{1}$ and $\beta_{2}$ at time step *t*, respectively.

The correction values of the parameter $\Delta w_{t}$ and $\Delta b_{t}$ are calculated based on the above deviations as follows:(24)$$\Delta w_{t}=-r\cdot {\hat{m}}_{w_{t}}/(\sqrt{{\hat{v}}_{w_{t}}}+\delta )$$
(25)$$\Delta b_{t}=-r\cdot {\hat{m}}_{b_{t}}/(\sqrt{{\hat{v}}_{b_{t}}}+\delta )$$
(26)$$w_{t}=w_{t-1}+\Delta w_{t}$$
(27)$$b_{t}=b_{t-1}+\Delta b_{t}$$

The parameter update process of the Adam algorithm acts on every element, thus the weight of every neuron of the proposed model is adjusted separately. For the training of the proposed model, the parameters were continuously updated by the Adam algorithm until the objective function error was less than convergence error ε.

### 4.2. Research on MWMC-CNN Algorithm

The above analysis is the forward propagation process of the proposed MWMC-CNN model, which can be divided into three parts: the moving window input part, the convolution and pooling part, and the fully connected output part. In the MWMC-CNN training process, the Adam algorithm is used to fine-tune parameters. This section gives a full description of the pseudo code of the MWMC-CNN model Algorithms 1 and 2. The details of the formulas in the algorithm are in Section 4.1.
**Algorithm 1** Energy consumption prediction algorithm of MWMC-CNN**Input A:**ID fan speed (XA1), EP fan speed (XA2), kiln average current (XA3), the amount of raw material (X1), the ECPC at historical moments (X2), and the CCPC at historical moments (X3)**Input B:**Primary cylinder temperature (XB1), decomposition furnace coal consumption (XB2), decomposition furnace temperature (XB3), kiln temperature (XB4),secondary air temperature (XB5), kiln head coal consumption (XB6), the amount of raw material (X1), the ECPC at historical moments (X2), and the CCPC at historical moments (X3).**Output:**The ECPC at future moments (y1), the CCPC at future moments (y2).**(Step 1) Initialization:****(Step 1.1) Normalization:**X←X^−X^minX^max−X^min**(Step 1.2) Moving Window Processing:**x(s×6)←(xA1,xA2,xA3,x1,x2,x3)Tx(s×9)←(xB1,xB2,xB3,xB4,xB5,xB6,x1,x2,x3)T**(Step 2) Convolution and pooling:****(Step 2.1) Convolution and activation:**A channel: ai((s−h+1)×6)←σrelu(wix(s×6)+bi),i=1,2,⋅⋅⋅nB channel: ai((s−h+1)×9)←σrelu(wix(s×9)+bi),i=1,2,⋅⋅⋅n**(Step 2.2) Pooling:**A channel: qi(((s−h+1)/k)×6)←favg-pooling(ai((s−h+1)×6)),i=1,2,⋅⋅⋅nB channel: qi(((s−h+1)/k)×9)←favg-pooling(ai((s−h+1)×9)),i=1,2,⋅⋅⋅nRepeat **Step 2.1–2.2** several times in each channel.**(Step 3) Full connection and output:****(Step 3.1) Full connection:**yz←∑u=1T1+T2σrelu(wzρu+bz),z=1,2,⋅⋅⋅T**(Step 3.2) Output:**yα←∑z=1T(yz),α=1,2

**Algorithm 2** Parameter adjustment algorithm of MWMC-CNNThe parameter update process of Adam algorithm for every element. **Objective function:**Minimize:J(w,b;x)←∑i=1n(yα(w,b;x)−y^)2
x←XA1(t−s+1)XA1(t−s+2)⋯XA1(t)XA2(t−a+1)XA2(t−a+2)⋯XA2(t)⋮⋮⋯⋮XB6(t−a+1)XB6(t−a+2)⋯XB6(t)T**Parameter Initialization:**r←0.001 (Learning rate)β1←0.9, β2←0.999 (Exponential decay rates of moment estimates)δ←10−8 (Constant)m0←0 (Initialize 1st moment vector)r←0 (Initialize 2nd moment vector)t←0 (Initialize time step)**while**
ε
**not converged do:**  t←t+1  xrandomL←{x(1),x(2),⋯,x(l),⋯,x(L)} (Randomly select *L* sets of data)  gwt=1L∇wt∑l=1LJ(wt−1;x(l)) (Calculate the gradient of wt)  gbt=1L∇bt∑l=1LJ(bt−1;x(l)) (Calculate the gradient of bt)  mwt←β1⋅mw(t−1)+(1−β1)⋅gwt (Update biased 1st moment estimate of wt)  mbt←β1⋅mb(t−1)+(1−β1)⋅gbt (Update biased 1st moment estimate of bt)  vwt←β2⋅vw(t−1)+(1−β2)⋅gwt (Update biased 2nd moment estimate of wt)  vbt←β2⋅vb(t−1)+(1−β2)⋅gbt (Update biased 2nd moment estimate of bt)  m^wt←mwt/(1−β1t) (Compute bias-corrected 1st moment estimate of wt)  m^bt←mbt/(1−β1t) (Compute bias-corrected 1st moment estimate of
bt)  v^wt←vwt/(1−β2t) (Compute bias-corrected 2nd moment estimate of wt)  v^bt←vbt/(1−β2t) (Compute bias-corrected 2nd moment estimate of bt)  Δwt←−r⋅m^wt/(v^wt+δ) (Calculate parameter update value of wt)  Δbt←−r⋅m^bt/(v^bt+δ) (Calculate parameter update value of bt)  wt←wt−1+Δwt (Update parameter wt)  bt←bt−1+Δbt (Update parameter bt)**end while****return**
wt, bt

## 5. Results

In this section, we used a MWMC-CNN model with two channels where each channel contained three convolutional layers and three pooling layers; a CNN model with three convolution layers and three pooling layers; a LSSVM model in which *p* = 0.03 and g = 0.01; and a LSTM model with two LSTM layers where each layer contains 48 hidden units to predict the electricity consumption and coal consumption of the cement calcination process. The experimental results were compared and analyzed to verify the superiority of the proposed model. In total, 12,500 datasets containing the electricity and coal consumption and cement calcination process variables stated in Section 4.1.1 sampled by sensors on the production line of a cement manufacturing enterprises in China were selected for the experiment, in which every variable in every group has one value, so every variable has 12,500 values. A total of 11,500 groups of all the data were used for model training, while the remaining 1000 groups were used as the test set.

The root mean square error (RMSE), the mean relative error (MRE), and the mean absolute error (MAE) are used as the indicators of predictive performance, whose formulae are respectively shown as:(28)$$RMSE=\sqrt{\frac{1}{m}\sum_{i=1}^{m}{\left({\hat{y}}_{i}-y_{i}\right)}^{2}}$$
(29)$$MRE=\frac{1}{m}\sum_{i=1}^{m}\left|\frac{{\hat{y}}_{i}-y_{i}}{y_{i}}\right|$$
(30)$$MAE=\frac{1}{m}\sum_{i=1}^{m}\left|{\hat{y}}_{i}-y_{i}\right|$$
where ${\hat{y}}_{i}$ is the sample value and $y_{i}$ is the predictive value. *m* is the size of the forecasting sample.

### 5.1. Parameter and Structure Adjustment Experiment of MWMC-CNN

The depth of the CNN and the size of the convolution kernel affect the performance of the proposed model. In order to find the parameters that make the proposed model perform better, we conducted a parameter adjustment experiment to compare the errors of the MWMC-CNN with different depths and different convolution kernel sizes under the actual cement calcination process data. The experimental results are shown in Figure 5.

We performed ten trials on MWMC-CNN with different sizes of the convolution kernel. All the above comparison experiments were optimized by the same optimization algorithm with the same parameters except for the depth and kernel size. The mean value of prediction errors of MWMC-CNN with six layers are represented by the dark-colored rectangles, and the line segment at the top of rectangle represents the variance of the experimental prediction error. The corresponding mean values and variances of the experimental prediction errors of MWMC-CNN with four layers are represented by light-colored rectangles and line segments above them, respectively.

As shown in Figure 5, the mean value and variance of experimental prediction error of MWMC-CNN with six convolution layers and a kernel size of 10-1 were the smallest. The above experimental results show that MWMC-CNN with six convolution layers is more suitable for feature extraction of the time series data of the cement calcination process. Using a convolution kernel size of 10-1 is beneficial to the improvement in the prediction performance of the proposed model.

### 5.2. Experiments Comparison of MWMC-CNN and Other Models

In order to show the intuitiveness of the experimental results, the MWMC-CNN, CNN, LSSVM, and LSTM were trained and tested under the same datasets, respectively. As a kind of highly competitive shallow network, LSSVM is widely used in industry field, so it is reasonable to choose LSSVM for the comparison. As a kind of time-dependent model, LSTM is designed to extract time series information and simulate long-term dependencies and short-term dependencies. The performance of time series information extraction and hidden feature mining of the proposed model can be verified by a comparison with LSTM and CNN, respectively. The experimental results are shown in Figure 6, Figure 7, Figure 8, Figure 9 and Figure 10. Figure 6 and Figure 7 indicate that all the models learned the characteristics of coal and electricity consumption synchronous prediction, which was the basis of the test. Figure 9 shows the loss curves of the models. Figure 8 and Figure 9 are the forecasting results, which demonstrate the superiority of the proposed model.

In order to evaluate whether LSSVM, CNN, LSTM, and MWMC-CNN learned the characteristics of training data, we carried out prediction experiments using the training dataset. The more overlapped the two lines, the more characteristics have been learned. The training result curves of the electricity and coal consumption of the four models are shown in Figure 6 and Figure 7, respectively, in which the blue line represents the original data samples while the red line represents the training output values of the corresponding model. The same termination condition of training for the four models was to reach a set error. It can be seen from the figures that the performances of the four models in the training stage were similar. The experiments indicate that all four prediction models grasped the characteristics of the training data.

Figure 8 shows the loss curves of CNN, LSTM, and MWMC-CNN. Due to the learning mechanism of hyperplane mapping, we did not draw the loss curve of LSSVM. It can be seen that all the losses converged to around 0. The fluctuation range of LSTM was the largest. The loss of MWMC-CNN was very close 0 and lower than that of CNN. In addition, the convergence speed of MWMC-CNN was faster than CNN.

To evaluate the actual prediction abilities of the four prediction models, we tested the models using the test dataset, which was different with training dataset. The test result curves of the electricity and coal consumption of the four models are shown in Figure 9 and Figure 10, respectively.

In Figure 9 and Figure 10, the blue line represents the original data samples while the red line represents the test output values of the corresponding model. Below every subgraph is the error curve of the corresponding model for the prediction index. It can be seen that the prediction curves of both the electricity and coal consumption of LSSVM were the worst and CNN was better than LSSVM, but worse than LSTM, which could match the actual value of electricity and coal consumption essentially but the precision was lower than MWMC-CNN. Through the comprehensive analysis of power and coal consumption synchronous prediction error curves of the four models, it can be concluded that the error curves of MWMC-CNN were the most stable among the four models for power and coal prediction synchronously. The experimental results verify the validity of the proposed model in this paper and demonstrate that the MWMC-CNN model has better generalization ability and higher precision for multi-energy index synchronous forecasting. The test errors of the four models are shown in Table 1.

As shown in Table 1, the minimum error was achieved when the convolution kernel size of MWMC-CNN with six convolution layers was 10-1. In all of the above results, deep networks such as CNN and LSTM performed better than the shallow network such as LSSVM in the time series prediction of the cement calcination process, which indicates that deep networks such as CNN are more suitable for the research in this paper. Compared with CNN and LSTM, MWMC-CNN had higher precision, the RMSE of MWMC-CNN was better than of 35.7% for CNN and 23.9% for LSTM; the MRE of MWMC-CNN was better than of 38.3% for CNN and 22.9% for LSTM; the MAE of MWMC-CNN was better than of 37.3% for CNN and 21.8% for LSTM. The results indicate that the proposed model has better time series feature extraction ability.

## 6. Conclusions

In this paper, we proposed a multi-channel convolutional neural network (MWMC-CNN) structure with a moving window based on the analysis of the mechanism of the cement calcination process and coupling relationship between the cement production process variables, which reduces the impact of time-varying delay in the cement calcination process on prediction accuracy while reducing the burden of industrial system modeling. The experimental results demonstrate the outstanding performance of the proposed model compared with the CNN model. In addition, according to different industrial mechanisms, the proposed model can be expanded to more output or input to deal with more complicated time series data, which provides a reference for the multi-output soft measurement modeling of the process industry and time series analysis of industrial big data. In our future studies, the proposed method will be expanded to predict the consumption of other kinds of energy or the energy consumption of other production processes, which provides a reference for energy scheduling and intelligent optimization control of the process industry.

## Figures and Tables

**Figure 1 sensors-21-04284-f001:**
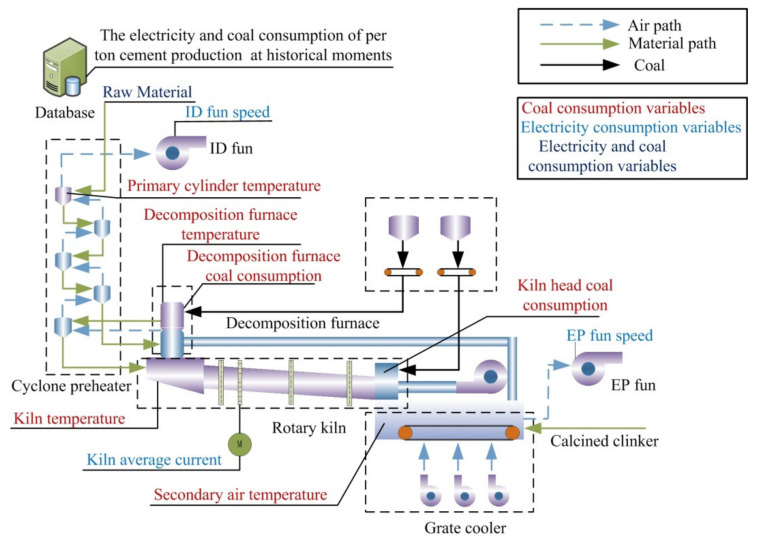
Calcination process of cement clinker.

**Figure 2 sensors-21-04284-f002:**
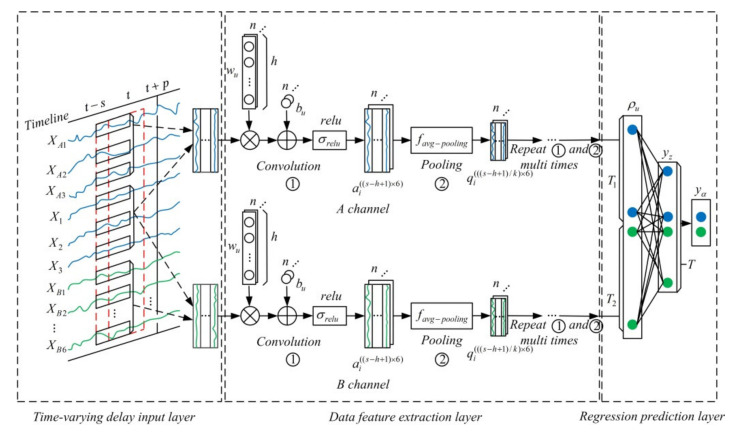
MWMC-CNN model structure.

**Figure 3 sensors-21-04284-f003:**
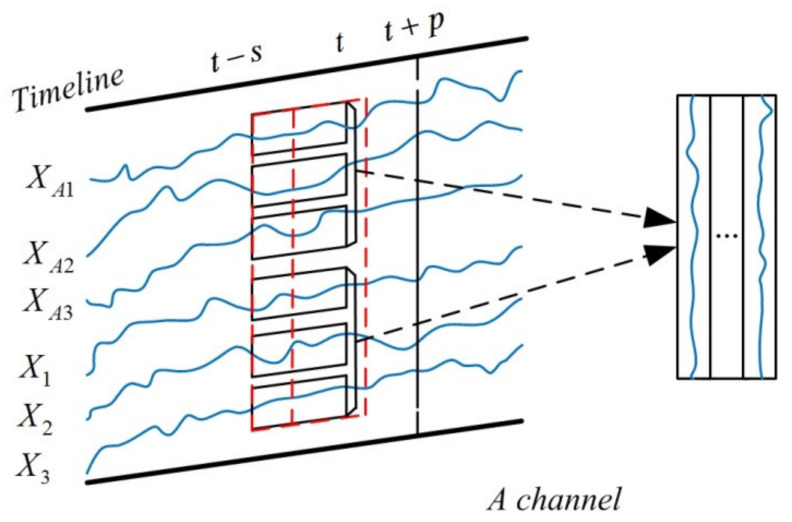
Time-varying delay input layer structure of the A channel of MWMC-CNN.

**Figure 4 sensors-21-04284-f004:**
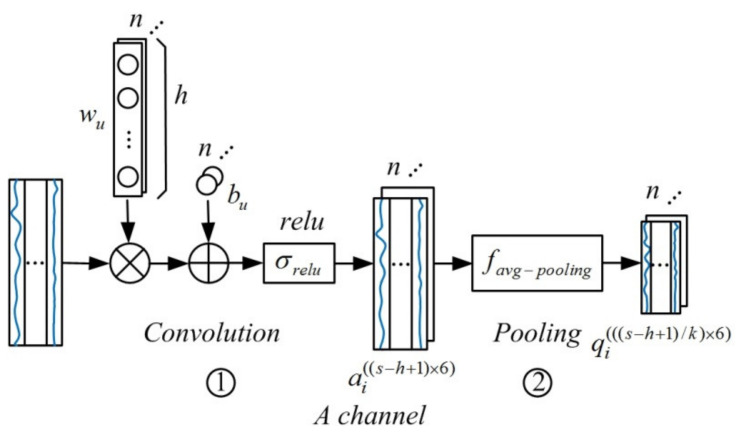
Data feature extraction process of MWMC-CNN.

**Figure 5 sensors-21-04284-f005:**
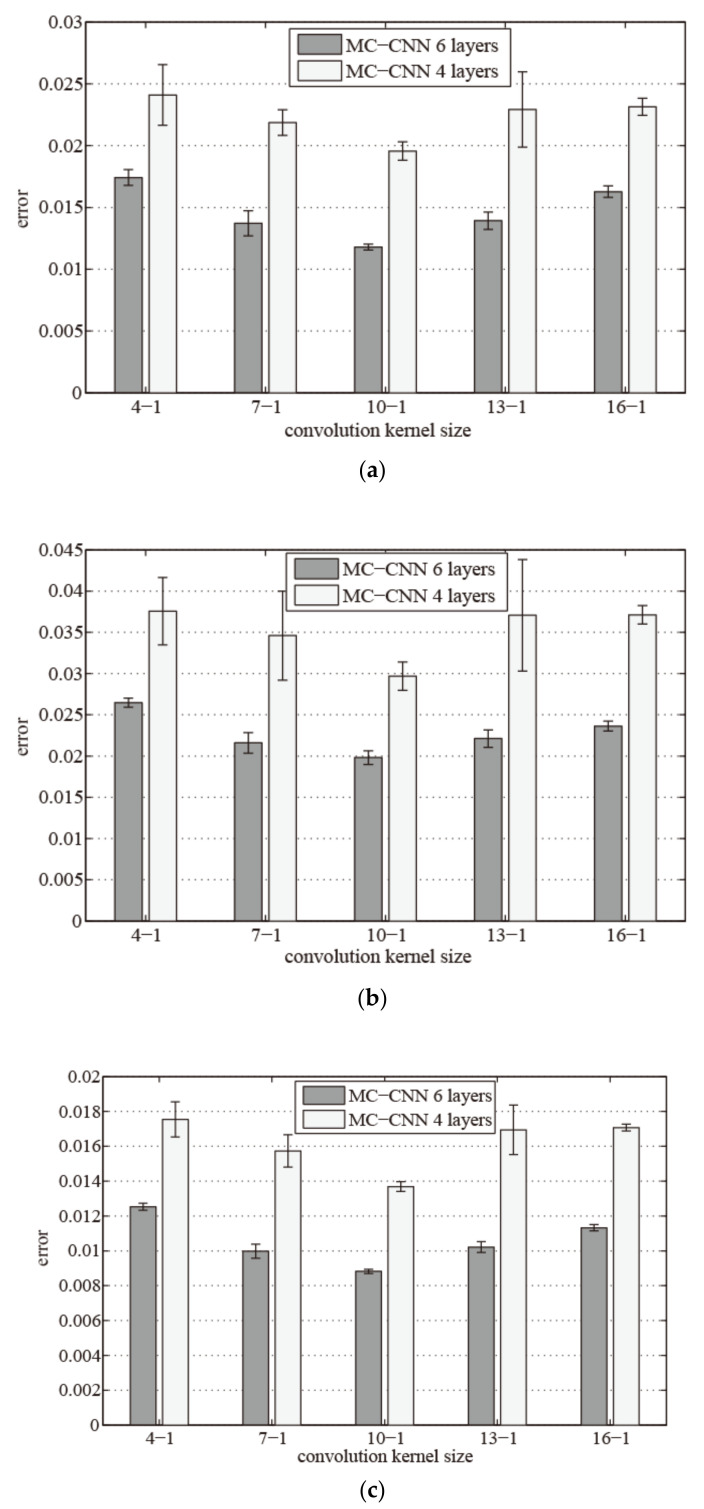
Error comparison of different structure MWMC-CNN. (**a**) RMSE comparison of different structure MWMC-CNN; (**b**) MRE comparison of different structure MWMC-CNN; and (**c**) MAE comparison of different structure MWMC-CNN.

**Figure 6 sensors-21-04284-f006:**
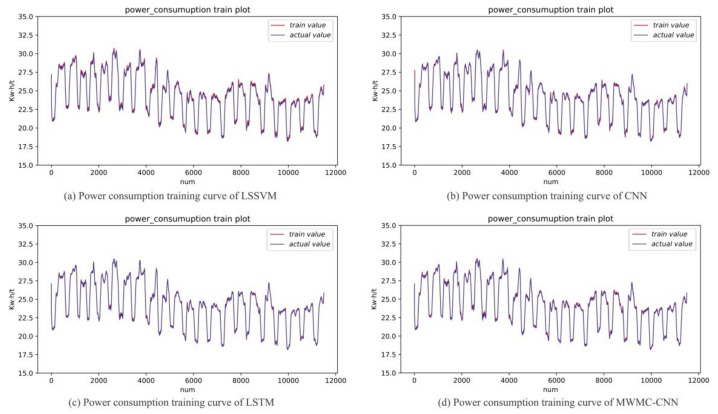
Electricity consumption training result curves of LSSVM, CNN, LSTM and MWMC-CNN.

**Figure 7 sensors-21-04284-f007:**
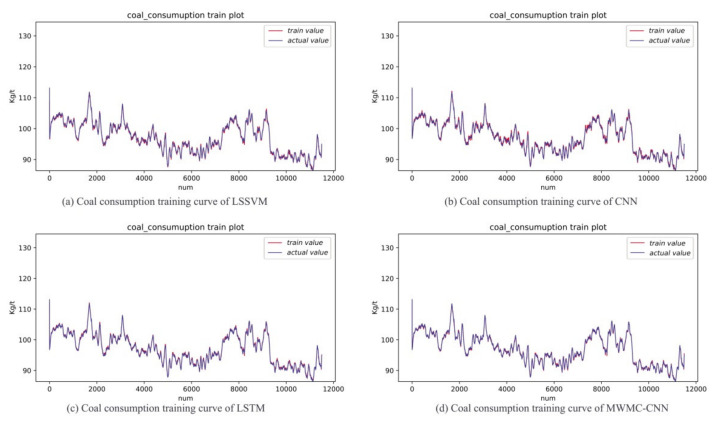
Coal consumption training result curves of LSSVM, CNN, LSTM and MWMC-CNN.

**Figure 8 sensors-21-04284-f008:**
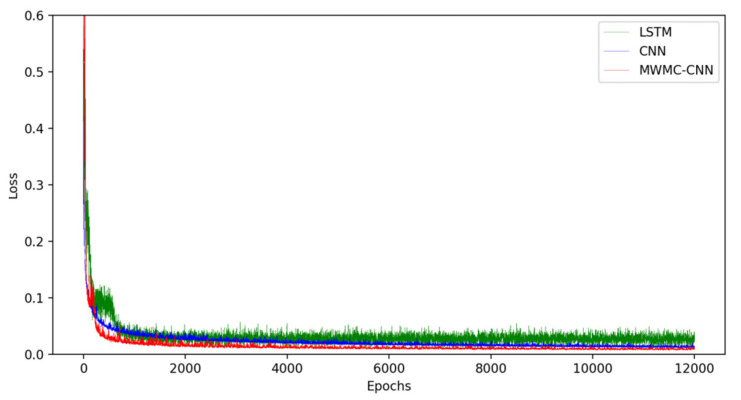
The loss curves of LSTM, CNN and MWMC-CNN.

**Figure 9 sensors-21-04284-f009:**
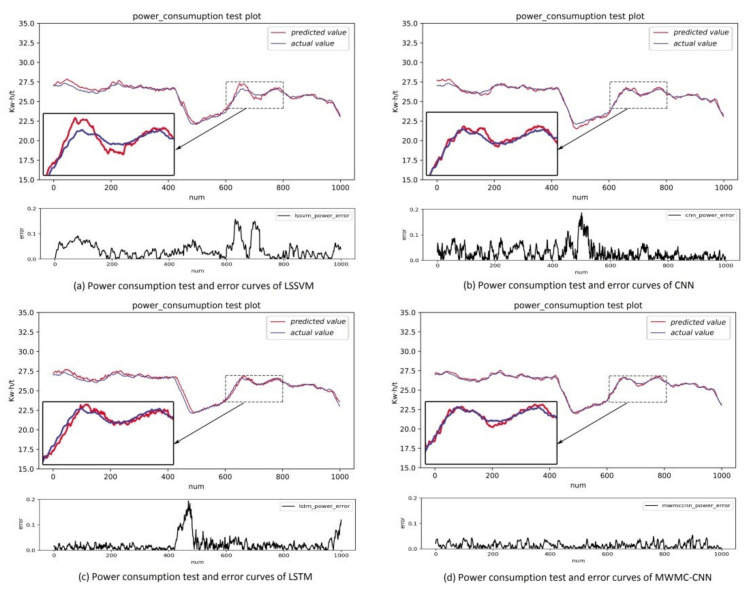
Electricity consumption test and error curves of LSSVM, CNN, LSTM and MWMC-CNN.

**Figure 10 sensors-21-04284-f010:**
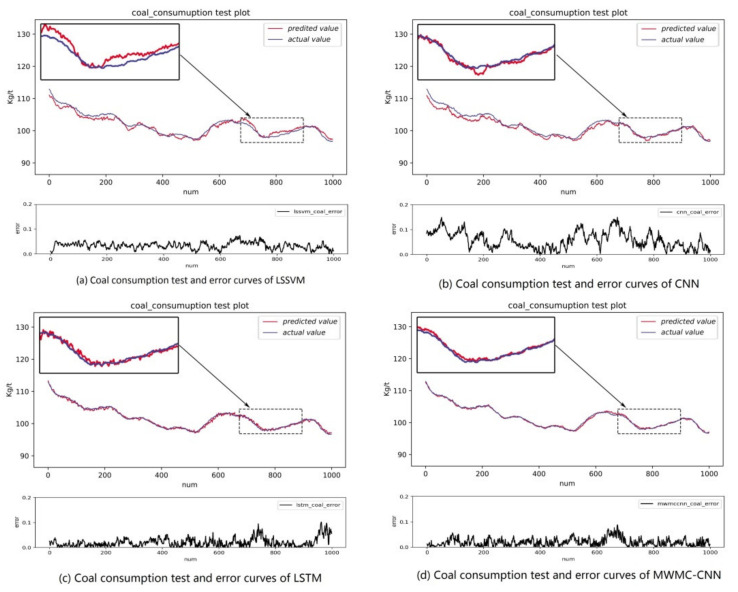
Coal consumption test and error curves of LSSVM, CNN, LSTM and MWMC-CNN.

**Table 1 sensors-21-04284-t001:** Error comparison of different models.

Model	Parameter	Prediction Error		
		RMSE	MRE	MAE
LSSVM	*p* = 0.03 g = 0.01	0.0172	0.0340	0.0132
CNN	4-1	0.0168	0.0284	0.0126
LSTM	48-48	0.0142	0.0227	0.0101
MWMC-CNN	10-1	0.0108	0.0175	0.0079

## Data Availability

Data sharing not applicable.

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
