# Peer review of "A Synchronous Prediction Model Based on Multi-Channel CNN with Moving Window for Coal and Electricity Consumption in Cement Calcination Process"

_sensors, 2021, doi:10.3390/s21134284_

Round 1

Reviewer 1 Report

The paper is concerned with a multi-channel convolutional neural network approach with moving window aiming to achieve an accurate prediction of multi energy consumption such as electricity and coal. The robustness of the proposed approach was evaluated by using raw data obtained from a cement plant. The paper is well written and structured making a good balance between the theoretical stuff needed to understand the authors’ proposed approach and critical discussion about the results achieved and, hence, for the reader is easy to follow the authors’ ideas. Thus, in the reviewer opinion the paper is recommendable for publication subject to minor changes according to the following comments:

  • Authors are encouraged to perform a final proofreading correcting typos, as for instance in pag: 4, line 182, “mage”;
  • Although the paper includes a good survey o published material related with the paper’s topic, there are missed relevant publications in this knowledge domain as for instance the following: “The daily and hourly energy consumption and load forecasting using artificial neural network method: a case study using a set of 93 households in Portugal”, Energy Procedia, 2014.

Author Response

Thank you for your comments, which we think will improve the manuscript. We have checked and modified the manuscript. All the modification have done in the manuscript using “track change” function and the illustrations point to point are shown as follow:

Point 1: Authors are encouraged to perform a final proofreading correcting typos, as for instance in pag: 4, line 182, “mage”;

Response 1: I’m sorry that there are some disadvantages are caused by typos or spell error. For example, “mage” in page 4 line 182 has been modified as “image”. Furthermore, we have checked the manuscript and modified other inappropriate contents, for example, “calcining” was changed to “calcination”.

In Introduction, “electricity consumption and coal consumption are measured mainly by sensor measurement and weighing” in Introduction was changed to “electricity consumption and coal consumption are measured mainly by sensors and weighing machines”, “the optimal solution of optimal control in production process may not be in solution space” was changed to “the optimal solution of optimal control of production process may not be in solution space”, “operator” in Introduction is changed to “operators”, “industry” in the third paragraph of Introduction was changed to “industrial”, “various data driven forecasting approaches have been proposed to predict or diagnose in industrial field” was changed to “various data driven forecasting approaches have been proposed for prediction or diagnosis in industrial field”, “power consumption of cement manufacturing is predicted using multiple non-linear regression algorithm [9] and electrical energy consumption of cement grinding process is forecasted by empirical mode decomposition based hybrid ensemble model [10]” was changed to “power consumption is predicted by multiple non-linear regression algorithm [9] and empirical mode decomposition based hybrid ensemble model [10]”, “solar radiationa” was changed to “solar radiation”, “explain” in sixth paragraph was changed to “explaining”, “in which the single index prediction strategy is firstly used to predict electricity and coal consumption, separately” was changed to “in which the single index prediction strategy is used to firstly predict electricity and coal consumption separately,”.

In Related Work, “create” was changed to “creates”, “industrial energy consumption” was changed to “energy consumption”, “Bianco used linear regression used to predict urban power consumption [18] and Catalina used to forecast thermal energy consumption of buildings [19]” was changed to “Bianco applied linear regression to predicting urban power consumption [18] and Catalina applied to forecasting thermal energy consumption of buildings [19]”, “it is also used to the prediction” was changed to “it is also used for the prediction”, “the difference may cause the loss of advantage of one-dimensional time series data” was changed to “the difference may causes the loss of advantages of one-dimensional time series data”, “caused by” was repetitive and one of them was deleted and “reference” was changed to “references” in the last paragraph.

In section 3, a verb “is” was added and “exchange” was changed to “exchanges” in paragraph 3.

In section 4, two “variable” in the fifth paragraph of 4.1.1 were changed to “variables”, “slid” was changed to “sliding”, “is” was changed to “are”, two “data of ” were deleted. In 4.1.2, “between” was changed to “among”, “each” was changed to “every”, an “of” was added, “is used to represent” was changed to “which represents” . In 4.1.3, two “represent” were changed to “represents”. In 4.1.4, “adjust” was changed to “adjusting”, “parameter” was changed to “parameters”.

In 5.2, “performance” was changed to “performances” and an “is” was changed to “are”.

In section 6, the “cement production” in the final of this section was changed to “process industry”.

Point 2: Although the paper includes a good survey o published material related with the paper’s topic, there are missed relevant publications in this knowledge domain as for instance the following: “The daily and hourly energy consumption and load forecasting using artificial neural network method: a case study using a set of 93 households in Portugal”, Energy Procedia, 2014.

Response 2: Thank you for your suggestion. There is a lack of the relevant investigations of electricity consumption forecasting in the manuscript. Thus, we have cited the literature: “The daily and hourly energy consumption and load forecasting using artificial neural network method: a case study using a set of 93 households in Portugal” was cited as the 36th citation and additional two literatures (citation 3 and 4) which are correlative with energy consumption prediction and the citations number are changed correspondingly.

Thank you sincerely again for your review of this manuscript.

Reviewer 2 Report

Strong points:

  • The design of the MWMC-CNN is clearly demonstrated.
  • Using paralleled CNNs to extract features from two input channels simultaneously can be an innovative design.

Weak points:

  • Too many typos and grammar errors make the essay hard to follow.
  • If the LSTM model was applied as a baseline model in section 5, the related works about the LSTM model should be included in section 2.
  • Please provide the detailed setups for your baseline models, especially LSTM and CNN models. From Table 1, looks like the kernel size of the baseline CNN model was 4 x 1. But how many convolutional layers did it contain? Additionally, the parameter of LSTM in Table 1 is incomprehensible. Did the LSTM baseline model contain 48 hidden states and 48 cell states? If yes, be specific.
  • When constructing the proposed CNN model, more justifications about the selection of the number of convolutional layers are required. The experiments only compared the performance of the proposed CNN model with 4 and 6 convolutional layers. What if the CNN model containing more convolutional layers? Would that increase the test accuracy?
  • Figure 6-7 are not clearly illustrated. The two lines are almost overlapped. Maybe draw the difference between the predicted value and actual value can be a better choice.

Author Response

Thank you for your comments, which we think will improve the manuscript. We have checked and modified the manuscript. All the modification have done in the manuscript using “track change” function and the illustrations point to point are shown as follow:

Point 1: Too many typos and grammar errors make the essay hard to follow.

Response 1: Thank you for your comment, we are sorry that there are some typo and grammar errors in the manuscript. We have checked it and modified some inappropriate contents, such as “calcining” was changed to “calcination”.

In Introduction, “electricity consumption and coal consumption are measured mainly by sensor measurement and weighing” was changed to “electricity consumption and coal consumption are measured mainly by sensors and weighing machines”, “the optimal solution of optimal control in production process may not be in solution space” was changed to “the optimal solution of optimal control of production process may not be in solution space”, “operator” in Introduction is changed to “operators”, “industry” in the third paragraph of Introduction was changed to “industrial”, “various data driven forecasting approaches have been proposed to predict or diagnose in industrial field” was changed to “various data driven forecasting approaches have been proposed for prediction or diagnosis in industrial field”, “power consumption of cement manufacturing is predicted using multiple non-linear regression algorithm [9] and electrical energy consumption of cement grinding process is forecasted by empirical mode decomposition based hybrid ensemble model [10]” was changed to “power consumption is predicted by multiple non-linear regression algorithm [9] and empirical mode decomposition based hybrid ensemble model [10]”, “solar radiation” was changed to “solar radiation”, “explain” in sixth paragraph was changed to “explaining”, “in which the single index prediction strategy is firstly used to predict electricity and coal consumption, separately” was changed to “in which the single index prediction strategy is used to firstly predict electricity and coal consumption separately,”.

In Related Work, “create” was changed to “creates”, “industrial energy consumption” was changed to “energy consumption”,  “Bianco used linear regression used to predict urban power consumption [18] and Catalina used to forecast thermal energy consumption of buildings [19]” was changed to “Bianco applied linear regression to predicting urban power consumption [18] and Catalina applied to forecasting thermal energy consumption of buildings [19]”, “it is also used to the prediction” was changed to “it is also used for the prediction”, “the difference may cause the loss of advantage of one-dimensional time series data” was changed to “the difference may causes the loss of advantages of one-dimensional time series data”, “caused by” was repetitive and one of them was deleted and “reference” was changed to “references” in the last paragraph.

In section 3, a verb “is” was added and “exchange” was changed to “exchanges” in paragraph 3.

In section 4, two “variable” in the fifth paragraph of 4.1.1 were changed to “variables”, “slid” was changed to “sliding”, “is” was changed to “are”, two “data of ” were deleted. In 4.1.2, “between” was changed to “among”, “each” was changed to “every”, an “of” was added, “is used to represent” was changed to “which represents”. In 4.1.3, two “represent” were changed to “represents”. In 4.1.4, “adjust” was changed to “adjusting”,  “parameter” was changed to “parameters”.

In 5.2, “performance” was changed to “performances” and an “is” was changed to “are”.

In section 6, the “cement production” in the final of this section was changed to “process industry”.

Point 2: If the LSTM model was applied as a baseline model in section 5, the related works about the LSTM model should be included in section 2.

Response 2: Thank you for your review, we realized that it is a defect when LSTM is used as a baseline model in section 5 while without included in section 2. Thus, we have add 5 citations in section2, which are all investigate energy consumption using LSTM. The increased related work is “LSTM is good at processing time series prediction because its special structure which is relevant to time factor. Thus, LSTM is applied to predicting energy consumption, for example gas consumption [40]. Besides, it is widely utilized for electricity consumption prediction, such as the energy consumption of housing [41] and commercial buildings [42], medium and long-term power forecasting [43] and electricity load forecasting in the electric power system [44]. Although all the above investigations have achieved the prediction of electricity consumption and used as references for electric departments or companies for decision-making of power production and dispatching, there is only one forecasting index which discrepant with the purpose of optimizing the industrial production process.”. And we found that although they focus on energy management and scheduling, their goals are provide references for energy sectors or companies to make decision for energy management or scheduling. It’s discrepant with the purpose of this manuscript, which is to provide references for factory for energy scheduling and optimizing production process to improve product quality, reduce energy consumption, reduce pollution emissions and protect the environment.

Point 3: Please provide the detailed setups for your baseline models, especially LSTM and CNN models. From Table 1, looks like the kernel size of the baseline CNN model was 4 x 1. But how many convolutional layers did it contain? Additionally, the parameter of LSTM in Table 1 is incomprehensible. Did the LSTM baseline model contain 48 hidden states and 48 cell states? If yes, be specific.

Response 3: Thank you for your comment. We are sorry that we didn’t give the structure of the models. The proposed MWMC-CNN model has two channels and each of them has 3 feature extraction  units in which a pooling layer behind a convolution layer. The baseline CNN model has 1 channel with 3 feature extraction units. The baseline LSTM model has two LSTM layers and each of them has 48 hidden units. In section 5 of the manuscript, the presentation of these models “In this section, we use MWMC-CNN, CNN, LSSVM and LSTM to predict the electricity consumption and coal consumption of cement calcination process.” has been changed to “In this section, we use a MWMC-CNN model with two channels and each of channels contains 3 convolutional layers and 3 pooling layers, a CNN model with 3 convolution layers and 3 pooling layers, an LSSVM model in which P=0.03 and g=0.01 and an LSTM model with two LSTM layers and each of layers contains 48 hidden units to predict the electricity consumption and coal consumption of cement calcination process.”.

Point 4: When constructing the proposed CNN model, more justifications about the selection of the number of convolutional layers are required. The experiments only compared the performance of the proposed CNN model with 4 and 6 convolutional layers. What if the CNN model containing more convolutional layers? Would that increase the test accuracy?

Response 4: Thank you for your comment. During our experiments, we found that over fitting may occur if  CNN containing more convolutional layers. It’s because our data scale is relatively small. In addition, considering our purpose is to investigate whether the proposed approach can overcome coupling relationship between electricity and coal consumption so that acquiring the guidance for cement to monitor and optimize the cement manufacturing process and formulate process control strategy. So as to improve the quality of cement, reduce energy consumption and pollutant emissions, and protect the environment. Thus, we don’t pay more attention to study how to gain the balance of more convolution layers and over fitting, maybe more accuracy prediction result can be got using MWMC-CNN model with more convolutional layers under the training of bigger scale data set. At that time, perhaps the choice of precision increment and computation cost will be another problem.

Point 5: Figure 6-7 are not clearly illustrated. The two lines are almost overlapped. Maybe draw the difference between the predicted value and actual value can be a better choice.

Response 5: Thank you for your review. Maybe we didn’t expound clearly the usage of figure 6-7. Figure 6 and 7 are illustrated using training data set to evaluate whether the prediction models have learned the characteristics. The more overlapped the two lines, the more characteristics have been learned. And difference between the predicted value and actual value are illustrated in figure 8 and 9. The corresponding contents have been added in 5.2 of the manuscript, such as “In order to evaluate whether LSSVM, CNN, LSTM and MWMC-CNN have learned the characteristics of training data. We carry out prediction experiments using training data set. The more overlapped the two lines, the more characteristics have been learned.” And “To evaluate the actual prediction abilities of the four prediction models, we test the models using test data set, which is different with training data set.”. in addition, “The test result curves of electricity and coal consumption of the four models are shown in Figure 8 and Figure 9.” Has been changed to “The test result curves of electricity and coal consumption of the four models are shown in Figure 8 and Figure 9, respectively.”

Thank you sincerely again for your review of this manuscript.

Round 2

Reviewer 2 Report

  1. Before abbreviation appearing for the first time, the complete words should be given. The full name of MWMC should be given. In Conclusion, the full name of MWMC-CNN appears again, the format should be unified.
  2. The author made sufficient literature review on the data driven prediction investigations in Related works on page 5. However, these backgrounds takes up too much space and are more reasonable to be put into Section 1, especially the contributions on page 5. They are your work, not 'related works', which should be listed in Section 1. The author should think twice about the whole article structure.
  1. Section 3 on is also tedious and lakes highlight.
  2. One table should be put into the same page.
  3. The author should explain more about the used data set, at least the author should tell your readers the size of each set and the number of each variable.
  4. In 5.2, the author claims that the superiority of the proposed model can be shown in Figure 6, Figure 7, Figure 8 and Figure 9. However, Figure 6 and Figure show similar results. The superiority cannot be concluded according to the author's explanation.
  5. The author should check your writing. For example, comma should be added when multiple figures appear together. In addition, it is better for the author to describe your work clearly in concise language.

Author Response

Response to Reviewer Comments

Thank you for your review which we think it helps to improve our manuscript. We have checked the manuscript again and made some modifications. The point to point details are as follow:

Point 1: Before abbreviation appearing for the first time, the complete words should be given. The full name of MWMC should be given. In Conclusion, the full name of MWMC-CNN appears again, the format should be unified.

Response 1: Thank you for your comment. We have checked the manuscript, the abbreviation MWMC only appears once by itself in the final of abstract, others always appears in the form of MWMC-CNN, which is in the parentheses behind its name “moving window and multi-channel convolutional neural networks”. The MWMC in the final of abstract means the structure of moving window and multi-channel, maybe there is a mistake. We have changed it to “the combination structure of  moving window, multi-channel with convolutional neural network”.

Point 2: The author made sufficient literature review on the data driven prediction investigations in Related works on page 5. However, these backgrounds takes up too much space and are more reasonable to be put into Section 1, especially the contributions on page 5. They are your work, not 'related works', which should be listed in Section 1. The author should think twice about the whole article structure.

Response 2: Thank you for your review. It’s more appropriate to place the characteristics part of the manuscript in section 1, we have did it. And the transitional sentence was changed to “The combination of moving window, multi-channel structure with CNN makes the two characteristics of this paper:”.

Point 3: Section 3 on is also tedious and lakes highlight.

Response 3: Thank you for your review. In section 3, we  described the  mechanism of cement calcination process. So the time delay of coal and electricity consumption and production process variables and the uncertain coupling relationship between coal and electricity consumption are illustrated, which are also the reasons we choosing the input process variables. Our thinking of the manuscript is like this. Firstly, we described the demand and significance of multi-energy index prediction synchronously. Then we illustrated and analysed the investigations of data driven method for energy forecasting. The investigate object was described in section 3, and the structure and algorithm of prediction model are described in section 4. Section 5 was experiments description. The final part was conclusions. Because our purpose is to investigate the method of synchronous forecasting of coal and electricity consumption in cement calcination process, so we think it’s necessary to describe the cause of the problem.

Point 4: One table should be put into the same page.

Response 4: Thank you for your review. I’m sorry that we didn’t treated the pseudo code as tables, we have changed them so that each of them is put in to the same page. And we have  emphasized their properties of pseudo code, and “This section gives a full description of the MWMC-CNN model algorithm” was changed to “This section gives a full description of pseudo code of the MWMC-CNN model algorithm”.

Point 5: The author should explain more about the used data set, at least the author should tell your readers the size of each set and the number of each variable.

Response 5: I’m sorry maybe we didn’t illustrated it clearly. In the experiment, every variable has 12500 values, in which 11500 groups are used as training data set and 1000 groups are test set. The description  “Totally 12500 sets data which contain electricity consumption and coal consumption and cement calcination process variable series data as stated in 4.1.1 collected by sensors on the production line of a cement manufacturing enterprises in China were selected for training, in which 11500 sets are used for model training, while the remaining 1000 sets are used as test sets.” in the manuscript was changed to “Totally 12500 sets data which contain electricity consumption and coal consumption and cement calcination process variable stated in 4.1.1 sampled by sensors on the production line of a cement manufacturing enterprises in China were selected for experiment, in which every variable in every group has one value, so every variable has 12500 values. 11500 groups of all the data are used for model training, while the remaining 1000 groups are used as test set.”

Point 6: In 5.2, the author claims that the superiority of the proposed model can be shown in Figure 6, Figure 7, Figure 8 and Figure 9. However, Figure 6 and Figure show similar results. The superiority cannot be concluded according to the author's explanation.

Response 6:  Thank you for your comment. This problem should have been revised last time, but we were negligent. I apologized for the negligence. The formulation in the manuscript “Comparative experiments training and test of the four models illustrate the superiority of the proposed model which are shown in Figure 6 Figure 7 Figure 8 and Figure 9.” has been changed to “the experiment results are shown in Figure 6, Figure 7, Figure 8 and Figure 9. Figure 6 and Figure 7 indicated that all the models have learned the characteristics of coal and electricity consumption synchronously prediction, which is the base of test. Figure 8 and Figure 9 are the forecasting results, which demonstrate the superiority of the proposed model.”.

Point 7: The author should check your writing. For example, comma should be added when multiple figures appear together. In addition, it is better for the author to describe your work clearly in concise language.

Response 7:  Thank you for your review. I’m sorry that some writing problems still exit in the manuscript. We have checked the manuscript and revised them. For example, “which means that the t-s~t period variables data in the , are adopted to predict the energy consumption indicators at t+p time stamps.” was changed to “which means that the t-s~t period variables data in the  are adopted to predict the energy consumption indicators at t+p time stamps.”. In addition, some missed space have been added and some upper case have been replaced by lower case, such as “Kiln” was changed to “kiln”, “The” was changed to “the”, “Decomposition” was changed to “decomposition” and “Secondary” was changed to “secondary”.

Thank you for your review again and apologize again. The revisions mode may cause some format problems again, such as figure and its title are not in the same page, which can be solved when the revisions are accepted. Thank you for your review sincerely again。